# MR Virtual Biopsy of Solid Renal Masses: An Algorithmic Approach

**DOI:** 10.3390/cancers15102799

**Published:** 2023-05-17

**Authors:** Stephane Chartier, Hina Arif-Tiwari

**Affiliations:** Department of Medical Imaging, College of Medicine, The University of Arizona, Tucson, AZ 85724, USA; chartier@arizona.edu

**Keywords:** MRI, image-guided biopsy, solid renal mass, oncocytoma, angiomyolipoma, renal cell carcinoma, lymphoma, transitional cell carcinoma

## Abstract

**Simple Summary:**

The incidence of solid renal masses has been steadily increasing over the past couple of decades. Most renal masses are now incidental findings on ultrasound or cross-sectional imaging studies performed for unrelated complaints. Our aim for this review is to provide a comprehensive discussion of the typical magnetic resonance (MR) findings for solid renal masses and create an algorithmic approach that could guide future imagers. With its low risk of harm to patients and ability to characterize multiple tissue parameters, MR imaging has supplanted computed tomography (CT) imaging as the most accurate way to “virtually biopsy” a small renal tumor. Modern-day MR protocols can provide clinicians a robust toolset for differentiating tumor subtypes, prediction of tumor histology, and pre-operative planning.

**Abstract:**

Between 1983 and 2002, the incidence of solid renal tumors increased from 7.1 to 10.8 cases per 100,000. This is in large part due to the increase in the volume of ultrasound and cross-sectional imaging, although a majority of solid renal tumors are still found incidentally. Ultrasound and computed tomography (CT) have been the mainstay of renal mass screening and diagnosis but recent advances in magnetic resonance (MR) technology have made this the optimal choice when diagnosing and staging renal tumors. Our purpose in writing this review is to survey the modern MR imaging approach to benign and malignant solid renal tumors, consolidate the various imaging findings into an easy-to-read reference, and provide an imaging-based, algorithmic approach to renal mass characterization for clinicians. MR is at the forefront of renal mass characterization, surpassing ultrasound and CT in its ability to describe multiple tissue parameters and predict tumor biology. Cutting-edge MR protocols and the integration of diagnostic algorithms can improve patient outcomes, allowing the imager to narrow the differential and better guide oncologic and surgical management.

## 1. Introduction

The incidence of solid renal tumors has been increasing steadily in recent decades, with the rate at 7.1 to 10.8 cases per 100,000 between 1983 and 2002 [1,2]. Mortality from this same time period presented a paradoxical finding; the increased rate of diagnosis did not improve mortality, which increased from 1.5 to 6.5 deaths per 100 [3]. However, a more recent analysis of renal tumor staging and outcomes presents a more optimistic picture. Patel et al. showed that the rising incidence of renal cell carcinoma (RCC) has been accompanied by a steady migration of initial tumor staging toward early-stage (stage I) cancer (44% in 2004 to 48% in 2015), a decrease in diagnosis of stages III and IV (stage III: 10% in 2004 to 7% in 2015; stage IV: 13% in 2004 to 11% in 2015), and an increase in five-year overall survival from 67.9% in 2004 to 72.3% in 2010 [4]. Interestingly, the advances in survival were seen exclusively in patients with advanced RCC (stage III and IV), while survival rates for lower stages have essentially leveled off at the time of publishing in 2015. This last point suggests that additional gains in survival are largely based on improvements in treatment options as new systemic therapies were approved for advanced kidney cancer [5,6,7], while treatment of primary tumors with nephrectomy is still utilized as the mainstay therapy in low-stage kidney cancer [8]. Thus, as survivability and treatment options for the localized disease have plateaued, advancements in imaging to better predict histological subtypes and grades may further contribute to improved survivability in the coming years.

Historically, renal masses were discovered in the exam room in patients presenting with the classic triad of hematuria, pain, and a palpable mass. Furthermore, the pre-operative diagnosis of these was generally obtained by percutaneous biopsy [9]. Today, most renal masses are detected as incidental findings on cross-sectional imaging studies performed for unrelated complaints, and are often initially described as small (≤4 cm) with T1a staging [10]. A vast majority of these tumors either grow slowly or demonstrate no detectable growth over time [11]. When a renal mass is detected, the first step is to determine whether it is a benign cyst or a higher-risk solid tumor, since up to 90% of resected solid renal tumors are malignant [12]. The Bosniak classification system divides cystic renal lesions into five types based on their contrast-enhanced computed tomography (CT) imaging findings [13]. However, solid renal tumor subtypes are varied and can be placed on a large spectrum in terms of aggressiveness, ranging from benign angiomyolipomas (AML) to high-grade RCCs. Thus, significant efforts have been made over the years to establish specific diagnostic criteria and management options for solid renal tumors [14].

Advancements in high-resolution cross-sectional imaging with CT, magnetic resonance (MR) imaging, and contrast-enhanced ultrasound (CEUS) have allowed imagers to better characterize renal masses and determine appropriate management. Of these, MR imaging has been particularly effective and has provided a toolset that allows the reader to differentiate renal tumor subtypes, establish tumor grading, provide pre-operative planning, and even predict histologic subtypes in some cases.

In this paper, we will summarize and discuss current diagnostic modalities, review key MR imaging characteristics for various solid renal tumor subtypes, and explore the future directions of MR in advancing care for solid renal tumors.

## 2. Imaging Modalities

The widespread use of cross-sectional imaging has led to an increased incidence of solid renal masses and an increase in documented subtypes of renal tumors [15]. Additionally, accurate characterization of these masses is critical for determining post-diagnosis screening and the optimal therapeutic approach to reduce patient morbidity and mortality. Advancements in imaging have improved the radiologist’s ability to differentiate between benign and potentially malignant masses. However, the wide array of imaging features, as well as overlapping features of benign and malignant tumors continue to pose a clinical challenge.

### 2.1. Ultrasound

Ultrasound (US) is often the first-line imaging option for detecting and characterizing renal masses. The primary advantages of this modality include the lack of ionizing radiation and the need for nephrotoxic computed tomography (CT) contrast agents. US is widely available and relatively low-cost compared to CT and magnetic resonance (MR) imaging. US is currently indicated in the workup of upper urinary tract symptoms and indeterminate renal masses, with an American College of Radiology (ACR) Appropriateness Criteria rating of 8 [16]. However, US is not indicated for the staging of renal cancer (ACR Appropriateness Criteria rating 3) [16].

Sonographic evidence of solid renal masses consists of a distortion of the normal tissue architecture with variations in echogenicity, size, and location. For the most part, solid renal masses are characterized as either completely solid or partially cystic, generally as a result of necrosis. Up to 70% of renal cell carcinomas (RCCs) ≤ 30 mm are seen as hyperechoic (relatively “bright”) to the normal cortex [17]. Size is a critical factor in determining the sensitivity of US in detecting tumors; 18% of tumors ≤ 20 mm and 21% of tumors 20–25 mm are not visible [18]. Color Doppler can help aid US in diagnosing and differentiating different types of solid renal masses. Jinzaki et al. demonstrated that with the use of color Doppler, the rate of correct diagnoses increased significantly with the use of color Doppler versus using grey scale alone. However, the paper demonstrated the limitations of US in that angiomyolipomas and particularly oncocytomas cannot be reliably differentiated from RCCs, which indicates the need for cross-sectional imaging for accurate characterization of solid renal masses [19].

### 2.2. Computed Tomography

CT is the most commonly used modality to evaluate and stage renal cancer. Due to its increased resolution over US and wider availability and decreased scan times compared to MR imaging, CT is the first choice of imaging in most cases. The limitations of CT are the use of ionizing radiation and the need for iodinated nephrotoxic contrast agents to adequately visualize and characterize renal masses. In general, a two-phase CT scan is optimal for renal mass characterization, consisting of a non-enhanced phase and a nephrogenic phase obtained approximately 90–120 s post-intravenous contrast administration [20]. However, when characterizing a renal mass for pre-procedure planning, an additional arterial phase is added to better depict vascular supply [20]. The nephrogenic phase yields homogenous enhancement of the normal renal parenchyma allowing for the detection of small renal lesions which may have been missed during the typical corticomedullary phase of a CT scan. The nephrogenic phase is significantly more sensitive in detecting renal masses, particularly those which are small (<11 mm) and medullary-based, compared to the corticomedullary phase. In fact, studies demonstrate a 50–60% increase in the detection rate of small renal masses in the nephrogenic vs. corticomedullary phases [21]. A third, excretory phase may prove useful in further characterizing renal cell carcinoma subtypes and assessing collecting system involvement by the tumor.

The use of intravenous iodinated contrast allows the radiologist to distinguish simple from complex cysts and solid renal tumors. The accepted threshold for “enhancing” is >20 Hounsfield units (HU) difference in attenuation between non-contrast and contrast-enhanced images [22]. Previously, the “pseudoenhancement” of cystic lesions has led to a misdiagnosis of solid renal mass; however, advancements in dual-energy CT have effectively eliminated this pitfall [23]. Not all renal tumors demonstrate a >20 HU difference on contrast-enhanced imaging and instead have equivocal enhancement of 10–20 HU. More recently, MR has supplanted CT as the primary modality for characterizing renal masses found incidentally on US or CT [24].

### 2.3. Magnetic Resonance

MR imaging has been gaining popularity in recent years as a problem-solving tool in the evaluation of renal masses described as indeterminate using US and CT. For this indication, the ACR Appropriateness criteria rates MR an 8 which is comparable to CT for the evaluation of renal masses and staging [16]. MR is particularly useful for patients who are unable to receive ionizing radiation or contrast agents. Limitations of MR include decreased availability of scanners and long acquisition times, incompatibility with certain metallic implants, and historic concerns related to the use of gadolinium-based contrast agents (GBCA). Most institutions use Group II gadolinium-based contrast agents such as gadobutrol (Gadavist) or gadobenate dimeglumine (MultiHance) which are considered safe agents with few, if any, unconfounded cases of nephrogenic systemic fibrosis and can be used in patients with chronic kidney disease stages of 4 or 5 [25]. MR imaging has significant soft tissue delineation compared to CT, thereby offering a more comprehensive evaluation of renal masses. In particular, MR allows the reader to better discriminate solid from cystic lesions when enhancement is equivocal on CT imaging (10–20 HU) [26]. Further discussion of renal, mass-specific MR protocols and MR-specific imaging features of renal tumor subtypes will follow in subsequent sections within this review.

## 3. Magnetic Resonance Imaging Approach

### Magnetic Resonance Imaging Protocol

Current MR protocols for imaging solid renal masses include multiple sequences allowing for comprehensive and systematic analysis of tumor phenotype. The Society of Abdominal Radiology Disease Focused Panel recently published a MR protocol as recommended by the 13 abdominal radiologists and 10 academic institutions they represent. The proposed protocol applies to a wide range of renal mass pathology including an indeterminate renal mass on US or CT, active surveillance of a known renal mass, post-ablation surveillance, and post-nephrectomy surveillance. The protocol recommends using gadolinium-based contrast material at a volume of 0.1 mL/kg body weight followed by a 10–20 mL saline flush. The Society of Abdominal Radiology lists four sequences as its core sequences: (1) two-dimensional (2D) T2-weighted single-shot fast spin echo; (2) 2D T1-weighted echo in- and out-of-phase; (3) T1-weighted pre-gadolinium enhanced fat-suppressed three-dimensional gradient-echo (FS 3D GRE); (4) T1-weighted dynamic post-gadolinium enhanced FS 3D GRE in arterial, venous, and delayed contrast phases. Generally, T2-weighted sequences allow for the detection and characterization of cystic lesions [27]. Additionally, T2-weighted images can be helpful in suggesting certain histology, which will be discussed further in subsequent sections. T1-weighted GRE in- and out-of-phase should be included for the thorough characterization of microscopic fat and detection of hemosiderin. The dynamic timing consists of the arterial phase (30 s), venous/corticomedullary phase (90–100 s), and the delayed/nephrogenic phase (180–210 s). This multi-phase enhancement protocol is useful in determining renal cell carcinoma (RCC) subtypes. A fourth, delayed phase may be obtained at 5–7 min post contrast to capture the excretory phase and image the ureters and urinary bladder. Additionally, diffusion-weighted images (DWI) may be obtained to further detect metastatic or nodal disease.

We present a systematic algorithm for the diagnosis and subtyping of small renal tumors: “virtual MR biopsy”; summarized in Figure 1.

An algorithmic approach to MR findings can be used to diagnose renal mass subtypes. The approach demonstrated above is based on a renal mass’s location of intralesional lipids, perfusion characteristics, and single-shot T2 signal intensity. Abbreviations: AML: angiomyolipoma; ccRCC: clear cell renal cell carcinoma; chRCC: chromophobe renal cell carcinoma; pRCC: papillary renal cell carcinoma; TCC: transitional cell carcinoma; MR: magnetic resonance; IR: inversion recovery; T1WI: T1 weighted image; ssT2WI: single-shot T2 weighted image; FS 3D GRE: fat-suppressed three-dimensional gradient-echo.

A summary of the imaging features of the discussed solid renal masses is shown in Table 1.

## 4. Benign Renal Tumors

### 4.1. Angiomyolipoma

Renal angiomyolipomas (AML) are the most common benign solid renal masses. As the name suggests, they are composed of dysplastic vasculature, smooth muscle, and lipid components. This tumor has a male to female ratio of 1:4 and is encountered in 2–6% of excised solid masses [28,29]. Approximately 80% of AMLs are sporadic with the remaining 20% having an association with tuberous sclerosis [30]. Of note, a macroscopic lipid is not evident on magnetic resonance (MR) imaging in 5% of AMLs; this subtype can be referred to as lipid-poor AML, minimal-lipid AML, and AML without visible lipid content [30,31].

On single-shot T2-weighted images (ssT2WI), classic AMLs demonstrate high-intensity signaling equivalent to surrounding mesenteric or retroperitoneal fat due to their fat content and will subsequently display low-intensity signaling on fat-suppressed images (e.g., spectral attenuated inversion recovery or India ink artifact) (Figure 2). It is important to note that the T2 signal of AMLs will be less than that of a true cyst, which is hyperintense on both non-fat-suppressed and fat-suppressed images due to internal fluid content. Observing the mass in both T1- and T2-weighted images (without and with fat saturation) is important to distinguish complex fluid (hemorrhagic and proteinaceous) content versus fat. Signal intensity on ssT2WI will be directly correlated with the AML’s overall lipid content (i.e., high lipid content will be hyperintense on ssT2WI and vice versa), although the T2 intensity of the AML will rarely exceed that of the surrounding mesenteric fat and will remain relatively hypointense compared to an adjacent cyst. Furthermore, the use of in- and out-of-phase T1-weighted imaging and the use of inversion recovery (IR) imaging can help localize microscopic/cytoplasmic and macroscopic lipid content, respectively [9]. Differentiating AMLs from clear cell RCC remains a diagnostic challenge due to overlapping imaging features. However, there is evidence suggesting that the presence of necrosis or intralesional calcifications favors a diagnosis of RCC versus a more benign lesion [32,33]. In these rare equivocal cases, image-guided biopsy should be obtained to exclude RCC.

Lipid-poor AMLs are less common than lipid-rich AMLs and continue to pose a challenge when it comes to imaging-only diagnosis. These masses are commonly hypointense on ssT2WI, similar to papillary RCCs (Figure 3) [34]. To distinguish the two, lipid-poor AMLs typically have a high level of arterial enhancement with subsequent washout, whereas papillary RCCs have low-level arterial and more pronounced delayed phase enhancement [34]. Lipid-poor AMLs can also be difficult to differentiate from clear cell RCCs, since both lesions can demonstrate diffuse signal loss on opposed-phase relative to in-phase images. Interestingly, a recent meta-analysis demonstrated the effectiveness of using a chemical shift signal intensity index to differentiate lipid-poor AMLs from RCCs in general. However, differentiating the subtypes of RCCs still poses a diagnostic challenge with chemical shift imaging [35]. A low T2 signal favors fat-poor AML over clear cell RCC, while a moderate to high-intensity signal would favor clear cell RCC. Ultimately, lipid-poor AMLs pose a diagnostic challenge for radiologists and may warrant a biopsy for definitive diagnosis when small (<3 cm).

### 4.2. Oncocytoma

Oncocytomas account for approximately 3–7% of solid renal masses and are considered benign [30,36]. This tumor is found more commonly in men over the age of 60. Importantly, oncocytomas and chromophobe RCCs share a common cellular origin, making this tumor difficult to distinguish on imaging alone. Classically, a central scar composed of fibrous and/or hyalinized connective tissue can be seen on gross pathology, although not all oncocytomas will have a scar. In addition, chromophobe RCCs may also present with central scarring [30,37].

Typical MR features include low signal intensity relative to renal parenchyma on T1-weighted images, high signal intensity relative to renal parenchyma on ssT2WI, general intralesional heterogeneity, and a contrast-enhancing central scar (Figure 3) [9].

The characteristics of scar enhancement are important when differentiating a benign oncocytoma and RCC. Oncocytomas demonstrate delayed scar enhancement in 74% of cases while this finding is present in only 12% of RCCs [29]. Relatedly, segmental enhancement inversion has been widely described as a characteristic enhancement pattern of oncocytomas [9,29,30,38]. Rosenkrantz et al. observed this contrast pattern with MR imaging in 28.6% of oncocytomas and 13.3% of RCCs. They did not observe differentiating inversions within the tumor itself or with contrast enhancement within the central scar [38]. A subsequent study investigated whether the combination of dynamic contrast-enhanced T1-weighted and double-echo gradient-echo MR sequences could distinguish oncocytoma with high, central T2-weight signal intensity from RCC. They conclude that the absence of central area signal intensity inversion or the presence of signal drop-out on the chemical-shift sequence rules out the diagnosis of oncocytoma [39]. While oncocytomas can be distinguished from RCCs generally, the differentiation of this benign tumor from chromophobe RCC remains an ongoing challenge due to overlapping imaging features such as peripheral location within the renal cortex, circumscribed morphology, lack of perinephric fat/renal vein invasion, and segmental inversion after contrast administration [40].

## 5. Malignant Renal Tumors

### 5.1. Clear Cell Renal Cell Carcinoma

Clear cell renal carcinoma (ccRCC) is the most common subtype of RCC, accounting for 65–80% of cases [41]. A vast majority of cases (~95%) are sporadic, while the remaining 5% are associated with hereditary syndromes including von Hippel-Lindau disease and tuberous sclerosis [15]. Compared with other subtypes of RCC, the clear cell variant is associated with less favorable outcomes, with a 5-year survival rate of 44–69% [30].

MR imaging of typical ccRCCs demonstrates heterogenous high signal intensity on single-shot T2-weight images (ssT2WI) which is usually due to intratumor necrosis, cystic changes, or hemorrhage (Figure 4) [9,29].

A helpful distinguishing feature of ccRCCs from other RCC subtypes is the avid enhancement in the corticomedullary and nephrogenic phases due to the hypervascularity of the tumor [27]. However, this enhancement characteristic is not pathognomonic and may be seen in other hypervascular tumors such as AMLs and oncocytoma; a distinction can be made when considering the T2-weighted appearance in conjunction with the enhancement characteristics. Another typical feature in 60% of ccRCCs is the presence of intratumoral, microscopic lipids [9,30]. Microscopic fat will appear as intensity drop-out on T1-weighted dual echo opposed-phase images relative to in-phase images [9,30]. MR imaging of ccRCCs may also demonstrate necrosis in the central area of the tumor which appears as a low-intensity signal on T1- and high-intensity on T2-weighted images [42]. Furthermore, the presence of tumor necrosis correlates with higher-grade tumors [43]. Clear cell RCC tumors are known to invade surrounding blood vessels, most often the renal vein and inferior vena cava causing thrombosis. Multivariant models of MR imaging have shown that renal vein and retroperitoneal collateral tumoral thrombosis are predictive of high-grade ccRCC [44].

### 5.2. Papillary Renal Cell Carcinoma

Papillary renal cell carcinoma (pRCC) is the second most common subtype of RCC, accounting for 10–15% of cases [36,43]. In rare cases (4%), they can be bilateral and/or multifocal in 23% of cases [30]. It should be noted that multifocal pRCC does not correlate with this tumor’s grading or staging [45]. pRCCs generally have a better prognosis compared to ccRCC, having a 5-year survival rate of 90% and a lower incidence of metastasis [30,45].

Generally, MR imaging will demonstrate a small (<3 cm), well-circumscribed, homogenous, exophytic tumor [30]. As this tumor increases in size, there is a greater propensity for heterogeneity due to hemorrhage, necrosis, and calcification [46]. Type I pRCCs characteristically present with hypointense signal intensity on T2-weighted images relative to normal renal cortex and demonstrate progressive enhancement (Figure 5) [30].

Type II tumors are generally larger, have indistinct margins, and have heterogenous T2-weighted signal and enhancement due to increased vascularity and necrosis [1]. Additionally, pRCCs will have lower signal intensity compared to normal renal parenchyma on ssT2WI secondary to hemosiderin deposits [9,34]. The presence of hemosiderin is noted by a relative drop of signal intensity on T1-weighted dual echo in-phase versus out-of-phase images, unlike ccRCCs which show drop-out on out-of-phase images due to microscopic/cytoplasmic lipid content. Lipid-poor AMLs can also appear hypointense on ssT2WI, similar to pRCCs. However, lipid-poor AMLs will have a high level of enhancement shortly after contrast administration while pRCCs will have progressive delayed enhancement, thereby making this distinction remarkable [9].

### 5.3. Chromophobe Renal Cell Carcinoma

Chromophobe renal cell carcinomas (chRCCs) represent approximately 5% of renal tumors, making chRCCs the third most common subtype of RCC behind the clear cell and papillary RCC tumors [43]. Similar to pRCCs, chRCCs are associated with a better prognosis and have a 5-year survival rate of 78–94% [47]. ChRCCs are found in adults and have an equal predisposition for males and females [1]. Of note, this tumor type is associated with Birt–Hogg–Dubé syndrome, an autosomal dominant disorder caused by mutations in the folliculin gene; oncocytomas are also associated with this syndrome and are more commonly found versus chRCCs [1].

Given the similarities between chRCC and oncocytoma, differentiation in MR imaging is challenging. ChRCCs are generally peripherally located, well circumscribed, homogenous, and lacking in perinephric fat or renal vein invasion [9]. Cystic changes and necrosis are uncommon even in large chRCCs [48]. Although chRCCs tend to have low to intermediate T2-weighted signal intensity, this finding can vary significantly [30]. Gadolinium administration will result in intermediate, delayed enhancement of chRCCs, less than ccRCCs but greater than pRCCs [1,30]. Additional enhancement patterns include a central, stellate scar (30–40% of cases) and “spoke wheel” enhancement, although these features may also be present in oncocytomas (Figure 6) [38].

## 6. Rare Renal Tumors

### 6.1. Lymphoma

The most common cause of renal lymphoma is secondary spread of non-Hodgkin disease, with a prevalence of 50% on autopsy; primary renal lymphoma is rare (<1%) [49,50]. Renal lymphoma can present as a solitary lesion, multiple masses, and/or retroperitoneal or perirenal disease. Of note, perinephric disease is a strong predictor for secondary renal lymphoma. Most patients (60%) present with multiple, 1–3 cm masses; solitary lesions only occur in 10–20% of patients [26]. On magnetic resonance (MR) imaging, renal lymphoma typically has low to intermediate signal intensity on T1- and T2-weighted images (Figure 7) [49]. A high-intensity, heterogenous T2-weighted signal may also be observed. Gadolinium contrast administration will reveal mild, delayed, and homogenous enhancement relative to the normal renal cortex [42,49].

### 6.2. Metastases

Metastatic disease of the kidney is relatively uncommon, occurring in 10% of extrarenal malignancy cases and usually in the late stages of the disease when widespread metastasis is already present [42]. Of those neoplasms that do metastasize to the kidney, bronchogenic carcinomas, gastrointestinal adenocarcinomas, and breast malignancies are the most common, with prostate, head and neck, and female genital tract being less common. Bilateral masses are found in 23% of cases and multiple masses are discovered in 30% of cases [51]. Given the range of tissue types that can metastasize to the kidneys, the MR findings are varied: small size, multifocal and/or bilateral, infiltrative or exophytic growth pattern, necrosis, hemorrhage, and calcifications have all been documented [1]. As such, there are no specific MR imaging characteristics that can aid in diagnosis. Rather, metastatic disease should be considered in the setting of multiple, atypical-appearing renal masses and a history of advanced, non-renal malignancy (Figure 8).

### 6.3. Transitional Cell Carcinoma

Transitional cell carcinomas (TCCs) of the renal pelvis are 50 times less likely than bladder TCCs, but only two to three times more common than TCCs of the ureter [52]. Renal pelvis RCCs are also less common compared to RCCs and makeup approximately 5–10% of renal tumors [53]. As with other TCCs, those in the renal pelvis are more prevalent in men and are typically diagnosed between 60–70 years of age [54]. Although a CT scan with delayed phase excretory images is the standard imaging modality for TCC, MR imaging still plays an important role. Urine provides excellent contrast on ssT2WI so filling defects and soft tissue masses are readily identifiable. When found, TCC masses in the renal pelvis display intermediate signal intensity on T1- and T2-weighted images, often isointense compared to normal renal parenchyma (Figure 9) [53]. The enhancement pattern using 3D GRE T1-weighted images typically shows delayed, heterogeneous enhancement.

## 7. The Role of Image-Guided Biopsy

Personalized decision making lies at the core of treatment protocols for tumors with variable risks of progression. As discussed, small renal cortical tumors are most often discovered incidentally and a substantial proportion are benign or indolent [3]. When imaging findings are indicative of a localized malignant small renal mass (clinical stage T1a), current guidelines recommend nephron-sparing surgery [55,56]. According to the American Urological Association, a percutaneous biopsy of a small renal mass can be considered when the mass is suspected to be hematologic, metastatic, inflammatory, or infectious; suspected cT1a renal tumors do not need to be routinely biopsied [55]. Direct complications of percutaneous biopsy are uncommon but can occur, especially in patients with uncontrolled hypertension. These complications include hematoma (the most common), clinically significant pain, gross hematuria, and hemorrhage (least common) [57]. Furthermore, when hemorrhage is present, additional intervention is often required in the form of trans-arterial embolization which carries its own risks.

In rare cases, a renal mass biopsy is performed when overlapping imaging features require histologic differentiation. For example, differentiating tumors with oncocytic origin (e.g., oncocytoma or chromophobe RCC) on imaging alone remains controversial due to overlapping imaging features [38]. Tissue sampling and special immunohistochemistry stains can be useful in this case. Cytokeratin 7 (CK7) stains cytoplasm. Oncocytomas generally show minimal CK7 staining, whereas chRCC is diffusely positive in a membranous distribution [37].

## 8. Recent Updates

### 8.1. Diffusion-Weighted Imaging

Diffusion-weighted imaging (DWI) for the characterization of renal masses has rapidly evolved in abdominal radiology in recent years. DWI has typically been investigated as a supplement to conventional T1- and T2-weighted images and dynamic contrast imaging. Solid malignancies generally have increased cellular density with intact cell membranes leading to increased restricted diffusion (and lower absolute diffusion coefficient (ADC) values). Investigators have studied the value of ADC measurements in predicting the histologic subtype of a renal mass in question and concluded that mapping ADC values can further differentiate low- from high-grade tumors [58,59]. Lassel et al. published a comprehensive meta-analysis comparing ADC values for benign tumors and malignant renal carcinomas. Their analysis showed carcinomas had significantly lower ADC values than benign renal tissue (1.61 ± 0.08 × 10^−3^ mm^2^/s vs. 2.10 ± 0.09 × 10^−3^ mm^2^/s; *p* < 0.0001), that oncocytomas had significantly higher ADC values than carcinoma (2.00 ± 0.08 × 10^−3^ mm^2^/s; *p* < 0.0001), and that urothelial tumors had the lowest ADC values (1.30 ± 0.11 × 10^−3^ mm^2^/s) [60]. More recent studies have confirmed these findings, solidifying ADC as a diagnostic tool for differentiating oncocytomas from similar-appearing malignant carcinomas [61,62].

### 8.2. Arterial Spin-Labeling

Arterial spin-labeling (ASL) is a functional MR imaging technique that characterizes the perfusion of renal masses without the use of intravenous contrast. Instead, the magnetic properties of water within free-flowing blood allow noninvasive quantification of tissue perfusion. This advanced technique was first pioneered in brain imaging and has been used to measure cerebral blood flow, detect cerebral vascular malformations, and characterize encephalitis [63]. Tumor vascularity is one of the most reliable features to help differentiate a renal mass from a benign tumor or normal renal parenchyma [64]. ASL is useful for monitoring tumor blood flow in patients undergoing anti-angiogenic therapy for RCC and for assessing the extent of tumor necrosis induced by these therapeutics [65]. More recently, ASL has made headway in helping surgeons plan and modify their surgical approach from radical to partial nephrectomy. If a partial nephrectomy is to be performed, it is necessary to know whether the perinephric fat is spared. RCCs generally do not have a true histologic capsule but instead possess a pseudocapsule composed of renal parenchyma and fibrous tissue [66]. Tumor sparing of the pseudocapsule, as characterized by multiplanar T2-weighted non-fat suppressed sequences, predicts nephron-sparing surgery [67]. Dynamic contrast-enhanced imaging has also been employed to characterize pseudocapsule involvement although this technique is imperfect as it cannot discriminate normal blood flow from tumor-related increased vascular permeability. Here, ASL is more specific as it is not affected by vessel permeability. Zhang et al. showed that using both T2-weighted and ASL sequences increased the specificity of identifying pseudocapsule involvement from 71.4% to 100% [68]. Additionally, one study showed that ASL has the ability to separate RCC subtypes based on perfusion level and to differentiate RCC from benign oncocytoma [69]. However, this technique is limited when detecting tumors with low perfusion such as papillary RCC [70]. ASL can also be used to assess treatment response in patients receiving vascular endothelial growth factor receptor (VEGFR) and/or tyrosine kinase inhibitors (TKIs) for metastatic RCC. Additionally, non-responders with early evidence of disease progression can also be identified and appropriately managed using this technique [71].

### 8.3. Magnetic Resonance Spectroscopy

Magnetic resonance spectroscopy (MRS) is a novel technique that allows tissues to be noninvasively interrogated for the presence and concentration of various metabolites. MRS has already been a proven method for the assessment of tissue characteristics in vivo and for the differentiation of brain, breast, and prostate tumors [72]. This technique is undergoing investigation as to whether it can be applied to renal tumors. Preliminary studies have shown its effectiveness in characterizing RCC when compared to normal renal tissue [73,74]. Renal MRS consists primarily of localized stimulated echo acquisition mode (STEAM) and point-resolved spectroscopy sequence (PRESS) to isolate various metabolites in normal parenchyma vs. a tumor [75]. Righi et al. highlighted the use of MRS in differentiating clear cell RCC from papillary RCC, although the study was limited in scope and pointed out several limitations of MRS including its increased sensitivity to respiration-related motion, voxel misregistration, and contamination from retroperitoneal fat [75]. Semilocalization by adiabatic selective refocusing (semi-LASER MRS) in MRI uses a single-shot MRS sequence to generate sharp voxel definition, outer volume suppression, and reduced chemical-shift displacement. This technique has been validated in the brain, prostate, and kidney since its recent development [76,77].

## 9. Newer Trends

### 9.1. Radiomics

Radiomics is the high-throughput extraction of minable, quantitative features from an image data set and the subsequent analysis of these data for clinical decision-making support [78]. Generally stated, the premise is that imaging data that is not perceptible to the human eye can convey meaningful information about tumor biology, behavior, and pathophysiology. When applying this novel concept to small renal masses, extracted data may allow for a finer distinction of nonmalignant versus malignant tumors, differentiating RCC subtypes, monitoring treatment response, and predicting prognosis in the metastatic setting [79,80]. Hoang et al. were one of the first groups to take techniques previously used to characterize pulmonary and colorectal tumors and apply them to differentiating RCC subtypes, with nearly 80% accuracy [81]. More recent studies have verified this initial study and report area under the curve (AUC) values ranging from 0.740 to 0.959 when differentiating clear cell, papillary, and chromophobe RCC subtypes [82,83,84]. Furthermore, MRI-based radiomics have proven to be superior to tumor size in predicting high-grade ccRCC disease with AUCs of 0.67 to 0.81 [85]. This advancement may allow improved patient selection when active surveillance is being considered for T1a/T1b renal masses (≤7 cm in size but limited to the kidney).

### 9.2. Radiogenomics

Another novel avenue of radiomics is identifying associations between imaging features and genomic signatures of small renal masses called radiogenomics. Radiogenomics is of particular interest as it would circumvent tissue sampling by characterizing surrogate imaging biomarkers that represent distinct tumor genotypes [86]. Currently, the vast majority of radiogenomics focuses on CT. MRI has the potential to greatly improve prediction models through its use of a variety of imaging sequences. Yin et al. have already demonstrated PET/MRI can differentiate molecular subtypes ccA and ccB for ccRCC, a robust predictor for localized and metastatic RCCs [87,88].

## 10. Conclusions

Imaging is a fundamental component of modern medicine. Its increased utilization has resulted in increased detection of small solid renal masses which encompass a wide spectrum of oncologic behavior. The ongoing advancements in imaging as well as oncologic and surgical treatment options have given rise to the “virtual biopsy” of solid renal masses. Ultrasound and CT are widely used for screening and diagnosis, especially for cystic lesions. However, when detected using these modalities, solid renal tumors are often described as indeterminate. Thus, MR has become the mainstay of solid renal mass imaging due to its low patient risk, high sensitivity, and ability to characterize a multitude of tissue parameters. The addition of cutting-edge protocols which incorporate diffusion imaging, arterial spin labeling, and MR spectroscopy has further bolstered MR’s role in determining the clinical significance of solid renal tumors. While considerable challenges remain in the definitive image characterization of these masses, further developments such as standardizing data collection, refining diagnostic algorithms, and integrating radiomics into reports have great potential to improve patient outcomes.

## Figures and Tables

**Figure 1 cancers-15-02799-f001:**
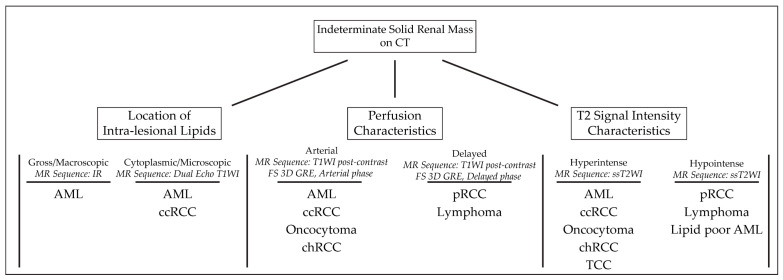
Algorithmic approach to solid benign and malignant renal tumors.

**Figure 2 cancers-15-02799-f002:**
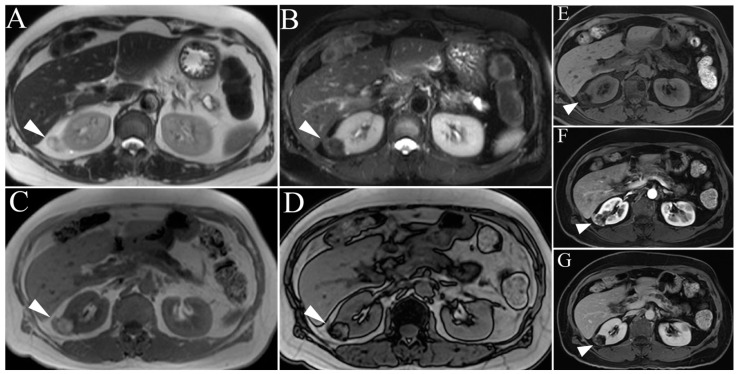
Angiomyolipoma. (**A**) Right renal lesion shows intermediate-to-heterogenous high signal (arrowhead) intensity on non-fat suppressed single shot T2WI, with hypointense appearance (arrowhead) on inversion recovery sequence due to macroscopic lipid content (**A**,**B**). T1 weighted in-phase (**C**) and out-of-phase (**D**) gradient echo images reveal characteristic India ink artifact around microscopic fat (arrowhead). (**E**) T1-weighted pre-gadolinium enhanced fat-suppressed three-dimensional gradient-echo (T1WI FS 3D GRE) image. (**F**) T1WI post-gadolinium FS 3D GRE, arterial phase image of the same angiomyolipoma. Note the marked increase in signal intensity during this phase compared to the non-contrast image (**A**). (**G**) T1WI post-gadolinium FS 3D GRE, delayed phase images show mild persistent contrast enhancement.

**Figure 3 cancers-15-02799-f003:**
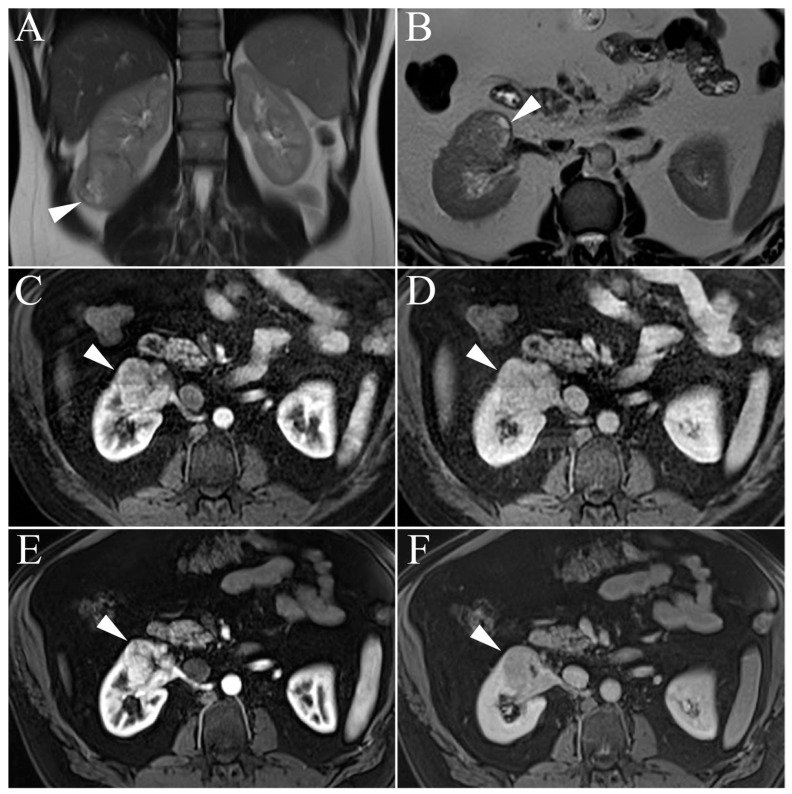
Oncocytoma. (**A**,**B**) Single-shot T2-weighted coronal and axial images show a right lower pole renal mass with increased signal within a central scar (arrowheads). (**C**,**D**) T1WI post-gadolinium enhanced fat-suppressed three-dimensional gradient-echo (T1WI FS 3D GRE) arterial and delayed phase images show relatively increased arterial enhancement of the oncocytoma tumor compared to the delayed phase (arrowheads). (**E**,**F**) The same right renal oncocytoma as (**C**,**D**) appears unchanged when compared with imaging from three years ago. Note identical MR features on arterial and delayed phase T1WI FS 3D GRE images (arrowheads), thereby favoring imaging diagnosis of benign oncocytoma.

**Figure 4 cancers-15-02799-f004:**
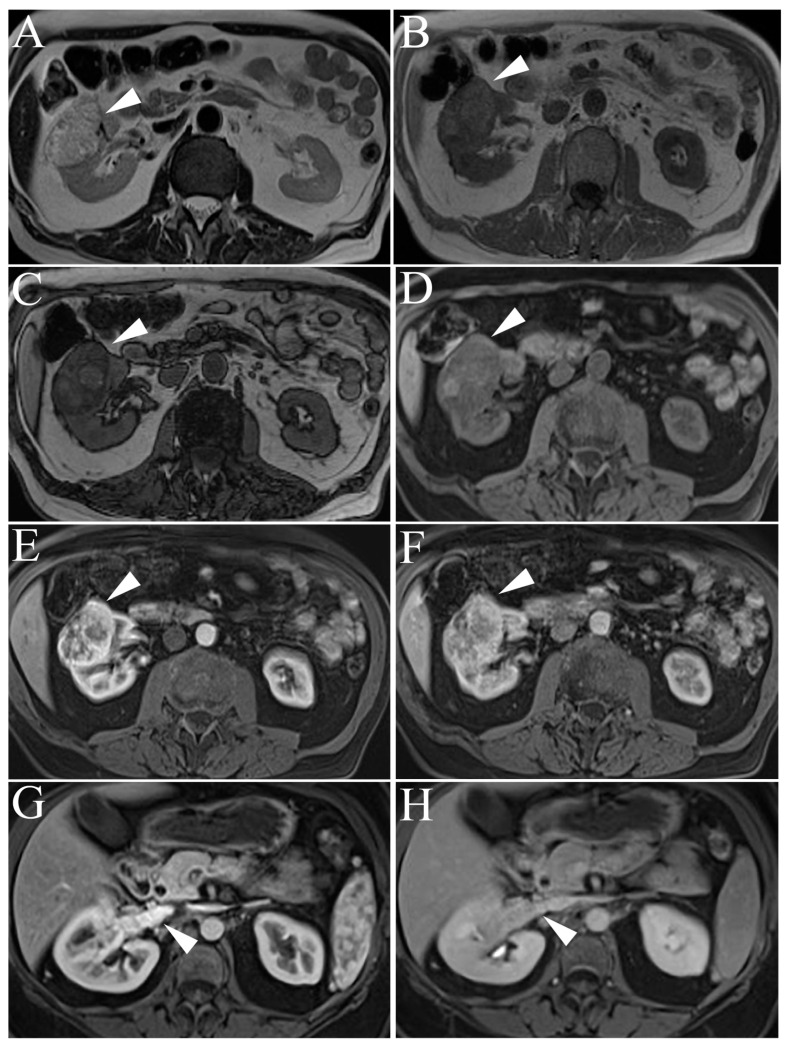
Clear cell renal cell carcinoma. (**A**) Single-shot T2-weighted image demonstrating high signal intensity of a clear cell renal cell carcinoma (ccRCC, arrowheads) in the anterior cortex of right kidney. (**B**) T1 weighted-image (T1WI), in-phase GRE image. (**C**) T1 out-of-phase GRE image shows signal “drop-out” secondary to microscopic/cytoplasmic intralesional lipid content. (**D**) T1-weighted pre-gadolinium enhanced fat-suppressed three-dimensional gradient-echo (T1WI FS 3D GRE) image. (**E**) T1WI post-gadolinium FS 3D GRE, arterial phase image. (**F**) T1WI post-gadolinium FS 3D GRE, delayed phase image. Note the increased arterial enhancement of the tumor as compared to delayed phase image. (**G**,**H**) Note the tumor thrombus in the right renal vein showing similar hypervascularity in arterial phase and relative washout in delayed phase images (arrowheads).

**Figure 5 cancers-15-02799-f005:**
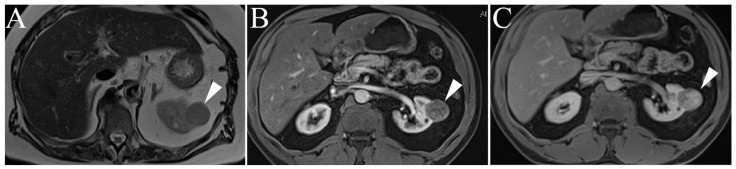
Papillary renal cell carcinoma. (**A**) Single-shot T2 weighted image shows left renal lesion with homogenous, hypointense signal (arrowhead) highly suggestive of papillary renal cell carcinoma (pRCC). (**B**,**C**) T1-weighted post-gadolinium enhanced fat-suppressed three-dimensional gradient-echo (T1WI FS 3D GRE), arterial phase image and delayed phase images reveal relative increased signal on delayed phase imaging compared to arterial phase (arrowheads).

**Figure 6 cancers-15-02799-f006:**
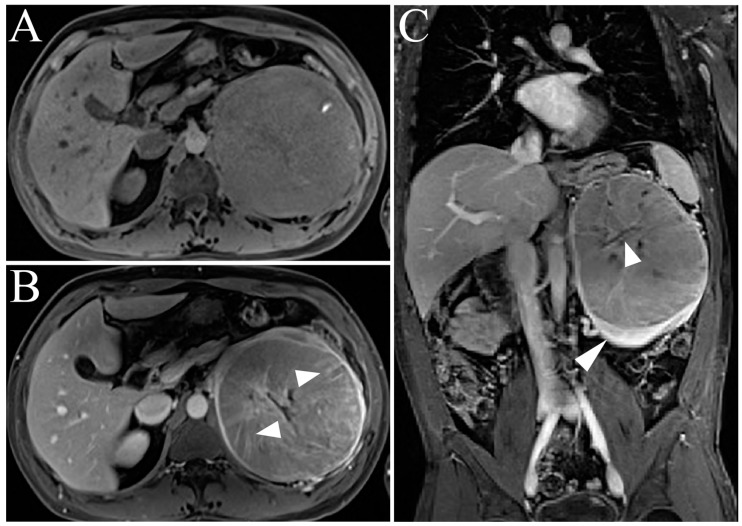
Chromophobe renal cell carcinoma. (**A**) Single-shot T2-weighted scan of a chromophobe renal cell carcinoma (chRCC). Note how the mass is well circumscribed with homogenous, low-intensity signal. (**B**) T1-weighted post-gadolinium enhanced fat-suppressed three-dimensional gradient-echo (T1WI FS 3D GRE), delayed phase image. Note the characteristic “spoke-wheel” enhancement pattern (arrowheads). (**C**) T1WI FS 3D GRE, delayed phase image demonstrating central scarring of the tumor (small arrowhead) and capsular enhancement (large arrowhead). Patient underwent nephrectomy; diagnosis of chRCC was confirmed on surgical pathology.

**Figure 7 cancers-15-02799-f007:**
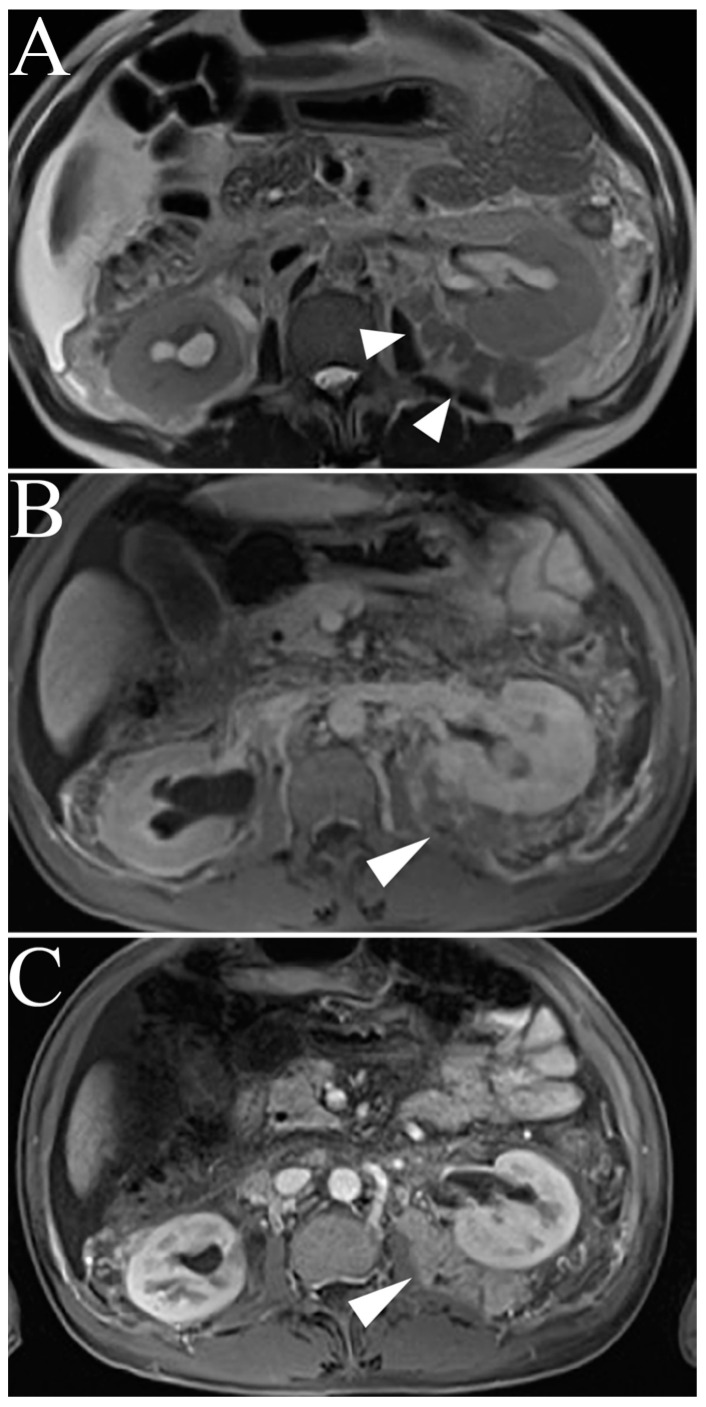
Lymphoma. (**A**) Single-shot T2-weighted image shows multiple perinephric masses with homogenous, low signal intensity (arrowhead). T1-weighted post-gadolinium enhanced fat-suppressed three-dimensional gradient-echo (FS 3D GRE) sequences in arterial (**B**) and delayed (**C**) phases shows progressive, persistent enhancement of perinephric soft tissue suggestive of secondary renal lymphoma (arrowheads).

**Figure 8 cancers-15-02799-f008:**
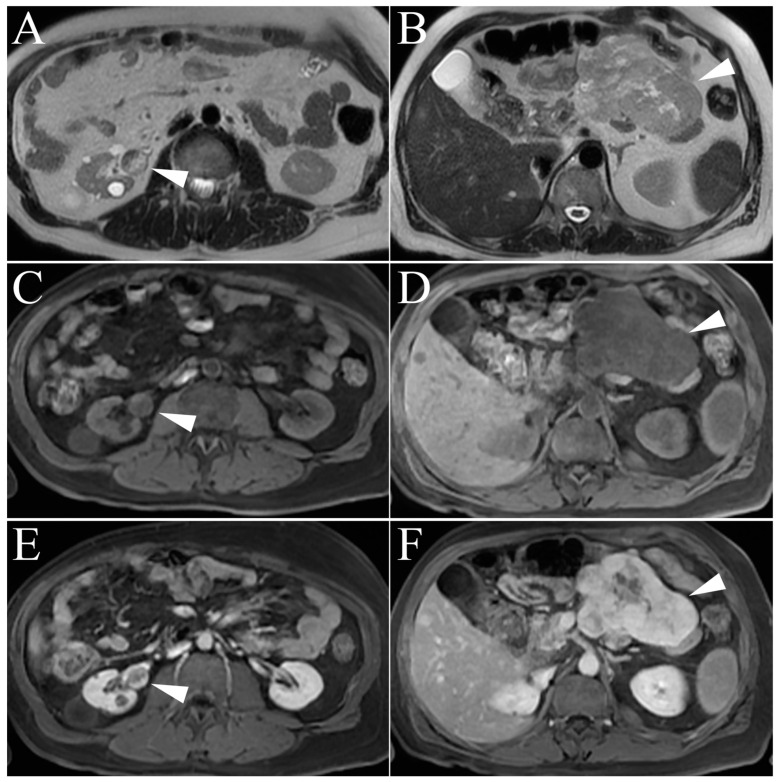
Metastases. (**A**,**B**) Single-shot T2-weighted image of stage IV leiomyosarcoma metastases in the right kidney (arrowhead) and biopsy-proved leiomyosarcoma of the mesentery (arrowhead). Note the similar heterogeneity of T2 signal of both primary tumor and new renal mass suggestive of metastasis. (**C**–**F**) T1-weighted pre- and post-gadolinium enhanced fat-suppressed three-dimensional gradient-echo (FS 3D GRE) images show similar pattern of intense, heterogenous enhancement in the delayed phase (arrowheads).

**Figure 9 cancers-15-02799-f009:**
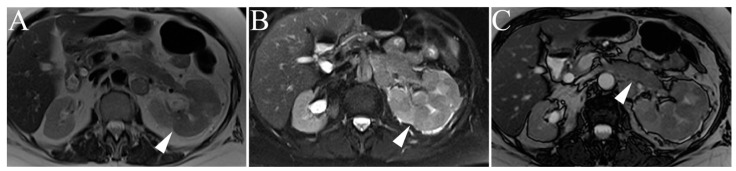
Transitional cell carcinoma. Transitional cell carcinoma. (**A**,**B**) Single-shot T2-weighted image, without and with fat saturation, shows transitional cell carcinoma in the left kidney. Note how the tumor conforms to the natural shape of the kidney and is nearly isointense to surrounding normal renal parenchyma (arrowheads). (**C**) True fast imaging with steady-state precession (TrueFISP) image demonstrating renal vein invasion by the tumor (arrowhead).

**Table 1 cancers-15-02799-t001:** Characteristic MR findings for benign and malignant renal tumors. Abbreviations: MR: magnetic resonance; GRE: gradient echo; T1W: T1-weighted; RCC: renal cell carcinoma; CC: cell carcinoma; IVC: inferior vena cava.

Type of Renal Masses	
Benign Masses	
Angiomyolipoma	High T2-intensity signal due to fat content. Low T2 on fat-suppressed images. Microscopic, intracytoplasmic fat made apparent with in- and out-of-phase GRE
Lipid-poor Angiomyolipoma	T2-hypointense Macroscopic fat and/or absence of fat High arterial enhancement with subsequent washout
Oncocytoma	T2-iso-to-hyperintense relative to normal parenchyma Central/eccentric T2-hyperintense scar Delayed enhancement of a central scar Segmental enhancement inversion pattern
Renal Cell Carcinomas	
Clear Cell RCC	Heterogenous, high T2-intensity Avid enhancement in corticomedullary and nephrogenic phases Microscopic fat as see on dual echo T1W in- and out-of-phase Invasion into surrounding vessels (esp. renal vein or IVC) Presence of necrosis or intralesional calcification
Type 1 Papillary RCC	T2-hypointense Uniform progressive delayed enhancement Well-circumscribed, homogenous, peripherally-located
Type 2 Papillary RCC	Heterogenous T2 signal intensity Heterogenous enhancement Larger with more indistinct margin vs versus Type 1 pRCC
Chromophobe RCC	Low to intermediate T2-intensity Intermediate, delayed enhancement Central, stellate scar with “spoke-wheel” enhancement pattern Peripheral, homogenous, well-circumscribed Mimics oncocytoma on imaging
Rare Renal Masses	
Renal Lymphoma	Low to intermediate T1 and T2 signal intensity Mild, delayed, homogenous enhancement Multiple 1–3 cm solitary masses
Metastasis	Varied presentation, usually identical to primary tumor Multiple, atypical renal masses History of advanced, non-renal malignancy
Transitional Cell Carcinoma	Intermediate T1 and T2 signal intensity Delayed, heterogenous enhancement Filling defects and soft masses when urine is present as contrast medium

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
