# Peer review of "MR Virtual Biopsy of Solid Renal Masses: An Algorithmic Approach"

_cancers, 2023, doi:10.3390/cancers15102799_

Round 1

Reviewer 1 Report

The authors systematically reviewed diagnostic imaging for renal tumors, and argued that the diagnostic accuracy of MRI, in particular, is comparable to that of tissue biopsy. Certainly, in clinical practice, MRI is useful for diagnosing renal tumors when CT alone is difficult to diagnose.

However, it seems that there are few cases where a biopsy is actually necessary. This is because if the tumor is localized and strongly suspected to be malignant, it is standard practice to remove it by surgery, whether it is papillary RCC or clear cell RCC.

However, it is necessary to improve the ability to distinguish between benign and malignant tumors in diagnostic imaging. The opinion of the authors, who argue that MRI is useful there, would be wonderful.

Situations that actually require a biopsy are often tumors with metastasis. At that time, the information obtained by biopsy is not only histological diagnosis, but also gene panel diagnosis using specimens, and discrimination of tumor malignancy, density of angiogenesis, presence or absence of sarcomatoid change components, etc. Considering that such results may provide hints for drug selection, virtual biopsy does not seem to be as good as tissue biopsy.

However, we sometimes face the disadvantages of actually biopsy. For example, you may experience pain, bleeding, and possibly bleeding requiring TAE. In order to reduce such disadvantages, it is necessary to make further studies so that image diagnosis approaches tissue diagnosis, as the authors say.

Author Response

Reviewer’s comment:

The authors systematically reviewed diagnostic imaging for renal tumors, and argued that the diagnostic accuracy of MRI, in particular, is comparable to that of tissue biopsy. Certainly, in clinical practice, MRI is useful for diagnosing renal tumors when CT alone is difficult to diagnose.

However, it seems that there are few cases where a biopsy is actually necessary. This is because if the tumor is localized and strongly suspected to be malignant, it is standard practice to remove it by surgery, whether it is papillary RCC or clear cell RCC.

However, it is necessary to improve the ability to distinguish between benign and malignant tumors in diagnostic imaging. The opinion of the authors, who argue that MRI is useful there, would be wonderful.

Situations that actually require a biopsy are often tumors with metastasis. At that time, the information obtained by biopsy is not only histological diagnosis, but also gene panel diagnosis using specimens, and discrimination of tumor malignancy, density of angiogenesis, presence or absence of sarcomatoid change components, etc. Considering that such results may provide hints for drug selection, virtual biopsy does not seem to be as good as tissue biopsy.

However, we sometimes face the disadvantages of actually biopsy. For example, you may experience pain, bleeding, and possibly bleeding requiring TAE. In order to reduce such disadvantages, it is necessary to make further studies so that image diagnosis approaches tissue diagnosis, as the authors say.

Authors response:

Thank you for your time to review our manuscript. We appreciate your thoughtful comment about need of biopsy for specific clinical situations and histological and genomic diagnosis. We have added a section on biopsy of frenal masses incorporating your recommendations.

Reviewer 2 Report

In this review article, the authors propose an algorithm for magnetic resonance (MR) based diagnosis and classification of solid renal tumors. The recommend MR protocols summarize published literature and are based on commonly and widely available clinical sequences. In addition, a short summary of sophisticated but less available methods (like arterial spin labelling and diffusion weighted imaging) and its contribution to characterize renal tumors is given. The features helping to identify the renal tumors is illustrate by numerous exemplary images. The manuscript is well written und well structured.

The review article may give radiologists a comprehensive summary for characterization of renal tumors by magnetic resonance and may therefore help to improve and standardize its clinical application.

Author Response

Reviewer’s comment:

In this review article, the authors propose an algorithm for magnetic resonance (MR) based diagnosis and classification of solid renal tumors. The recommend MR protocols summarize published literature and are based on commonly and widely available clinical sequences. In addition, a short summary of sophisticated but less available methods (like arterial spin labelling and diffusion weighted imaging) and its contribution to characterize renal tumors is given. The features helping to identify the renal tumors is illustrate by numerous exemplary images. The manuscript is well written und well structured.

The review article may give radiologists a comprehensive summary for characterization of renal tumors by magnetic resonance and may therefore help to improve and standardize its clinical application.

Authors response:

Thank you for reviewing our manuscript. Authors appreciate your feedback and comments.

Reviewer 3 Report

The authors present a nice attempt at summarizing the imaging characteristics of solid renal masses with a focus on MR imaging. However, the presentation of imaging features is confusing with many features discussed that are general features of all solid masses that would not be helpful in differentiating some of these lesions. Additionally, some claims in the tables and figures are a bit too bold for example figure 1 claiming presence of fat is AML without any nuance for RCC’s also sometimes containing fat which can have significant clinical consequences for the patient. The lesions presented as examples, are they all path proven because otherwise some definitely are not as clear based on imaging alone what the diagnosis is. This is a great idea to present a concise review of imaging features of solid renal masses however there are currently too many unclear and confusing points as well as description of imaging findings that isn’t entirely accurate. Hard to say that DWI is a future direction. DWI imaging is pretty widely used in body MRI and cancer imaging. Maybe some specific technique and ADC evaluations is what you meant to focus on.

Some mor discrete comments:

Line 37 – erroneous comma before presented

Intro – given advances in survival were seen with advanced rcc, how does that demonstrate importance of early detection and diagnosis? Have survival rates for early stage changed? How much of a change was there toward early stage disease/diagnosis ?

Line 51-52  - reference for most found as incidental findings?

Line 53 – reference for vast majority not growing or growing slowly?

Line 97/98 – that article cited actually states that RCC and oncocytomas cannot be differentiated from each other, a known problem

Line 105 – reference for only a 2 phase CT scan?

Line 118 – “equivocal “ enhancement is a more accurate term, not intermediate

Figure 1 – if it is a solid renal mass how can it be characterized as a simple cyst? What about t2 imaging? Also macroscopic fat does not mean AML by definition. Some RCC’s can have fat, too. For example: https://www.ncbi.nlm.nih.gov/pmc/articles/PMC7034758/  Presence of calcification can help differentiate but that can be difficult to ascertain on MRI.  Is this figure based on a specific paper? The radiologic characterization needs references if so.

Table 1 – simple cyst is also T2 hyperintense as a key imaging feature and should not be confused with a solid mass on imaging. AML would be T2 bright if the T2 imaging is not fat suppressed which is a common sequence in many institutions which you mention in the next line but how would the clinician realize which is fat suppressed and which isn’t? The key differentiation between AML and other renal massed is presence of fat but other renal masses can have fat too so it is not pathognomonic.

Line 185 – high intensity T2 usually implies something closer to CSF signal. The image shown is more moderate T2 signal. You can see the small cyst adjacent to the lesion with more high intense T2 signal for comparison

Line 189 – again high T2 signal implies closer to fluid signal

Line 191 – confusing statement re blood and proteinaceous fluid mimicking fat on t1. Depends on type of T1 and where it is fat satted which oftentimes it is and would therefore not be mimicking

Line 194 – that is more intermediate signal on T2

Line 201 – high signal intensity where? On F? would not agree

Line 208/209- so you are saying that all clear cell RCCs have fat in them? Reference?

Figure 3: C – cannot appreciate marked increase in signal as there are no precontrast images to compare to. Need either precontrast image or show subtraction images. E/F, if you want to show decrease in signal, should show ROI measurements because hard to appreciate any significant drop out. Was this lesion path proven fat poor AML?

Figure 4 – is this path proven oncocytoma? Because this figure can be of other types of solid renal mass as well based on the images provided

Figure 5 : C and D are different mass than A and B and then you go back to the original mass in E and F.. very confusing and provides only some imaging characteristics for each mass. Also B and E are the same image, both in phase T1. Should B have been a T1 fat sat? G and H – thrombus is still enhancing on delayed phase but less so, is that what you meant? Lack of enhancement would imply dark signal.

Line 296 – reference for progressive delayed enhancement? Papillary RCC tend to be hypoenhancing compared to renal parenchyma in general. https://pubs.rsna.org/doi/full/10.1148/rg.2017170039

Figure 7 C – arrowheads seem to be switched.

Line 259-261: reference?

Figure 9 – is this path proven because it really looks like a cyst on the limited provided imaging. What about DWI? Was the initial leiomyosarcoma so T2 hyperintense too?

Author Response

Reviewer’s comment:

The authors present a nice attempt at summarizing the imaging characteristics of solid renal masses with a focus on MR imaging. However, the presentation of imaging features is confusing with many features discussed that are general features of all solid masses that would not be helpful in differentiating some of these lesions. Additionally, some claims in the tables and figures are a bit too bold for example figure 1 claiming presence of fat is AML without any nuance for RCC’s also sometimes containing fat which can have significant clinical consequences for the patient. The lesions presented as examples, are they all path proven because otherwise some definitely are not as clear based on imaging alone what the diagnosis is. This is a great idea to present a concise review of imaging features of solid renal masses however there are currently too many unclear and confusing points as well as description of imaging findings that isn’t entirely accurate. Hard to say that DWI is a future direction. DWI imaging is pretty widely used in body MRI and cancer imaging. Maybe some specific technique and ADC evaluations is what you meant to focus on.

Authors response:

Authors are very appreciative of detailed review, analysis and all recommendations provided by reviewer 3.

We have made major changes to our manuscript based on your recommendations and comments. Certain figures have been removed since we did not have surgical pathology or biopsy to support imaging diagnosis. We have updated the flow chart to enhance clarity of MR findings.

Thank you for your comment about future directions- DWI/ADC, we have added appropriate comments and reference. Additionally, a new section on radiomics/radiogenomics has been added since it better aligns with the rapid advancement in the field of cancer imaging, particularly RCC.

Please see edits stated below based on your recommendation of specific areas in the manuscript.

Some more discrete comments:

Reviewer’s comment: Line 37 – erroneous comma before presented.

Authors response: Thank you for your feedback. We have removed the erroneous comma.

Reviewer’s comment: Intro – given advances in survival were seen with advanced rcc, how does that demonstrate importance of early detection and diagnosis? Have survival rates for early stage changed? How much of a change was there toward early stage disease/diagnosis ?

Authors response: Thank you for the comment; new text and references have been added with track changes for your review.

Reviewer’s comment:: Line 51-52  - reference for most found as incidental findings?

Authors response: A new reference is added. Thank you for the comment.

Reviewer’s comment: Line 53 – reference for vast majority not growing or growing slowly?

Authors response: A new reference is added. Thank you for the comment.

Reviewer’s comment: Line 97/98 – that article cited actually states that RCC and oncocytomas cannot be differentiated from each other, a known problem

Authors response: Thank you for the comment, we have updated the text stating the overlap between oncocytoma and chromophobe RCC.

Reviewer’s comment: Line 105 – reference for only a 2 phase CT scan?

Authors response:  We have added discussion for multiphase CT protocol with a reference

Reviewer’s comment: Line 118 – “equivocal “ enhancement is a more accurate term, not intermediate

Authors response: Thank you for the recommendation, we have changed intermediate to equivocal.

Reviewer’s comment: Figure 1 – if it is a solid renal mass how can it be characterized as a simple cyst? What about t2 imaging? Also macroscopic fat does not mean AML by definition. Some RCC’s can have fat, too. For example: https://www.ncbi.nlm.nih.gov/pmc/articles/PMC7034758/  Presence of calcification can help differentiate but that can be difficult to ascertain on MRI.  Is this figure based on a specific paper? The radiologic characterization needs references if so.

Authors response:  Thank you for your comment about differentiation of AML and cysts on to ssT2WI images. We have added appropriate details in text and new references, including the above reference.

Reviewer’s comment: Table 1 – simple cyst is also T2 hyperintense as a key imaging feature and should not be confused with a solid mass on imaging. AML would be T2 bright if the T2 imaging is not fat suppressed which is a common sequence in many institutions which you mention in the next line but how would the clinician realize which is fat suppressed and which isn’t? The key differentiation between AML and other renal massed is presence of fat but other renal masses can have fat too so it is not pathognomonic.

Authors response: Thank you for your comment. Table briefly discusses macroscopic and microscopic lipid content in angiomyolipoma and utilization of inversion recovery and dual echo T1WI for identification of lipid rich AML. These findings are also reviewed in the text under AML.

Reviewer’s comment: Line 185 – high intensity T2 usually implies something closer to CSF signal. The image shown is more moderate T2 signal. You can see the small cyst adjacent to the lesion with more high intense T2 signal for comparison

Authors response: We have added text details and reference for T2 hyperintensity in AML, and how it is different to cysts. Since AML contains internal lipid, T2 weighted signal is similar to retroperitoneal or mesenteric fat.

Reviewer’s comment: Line 189 – again high T2 signal implies closer to fluid signal

Authors response:  Thank you for your observation. We have added text details and reference for T2 hyperintensity in AML, and how it is different to cysts. Since AML contains internal lipid, T2 weighted signal is similar to retroperitoneal or mesenteric fat.

Reviewer’s comment: Line 191 – confusing statement re blood and proteinaceous fluid mimicking fat on t1. Depends on type of T1 and where it is fat satted which oftentimes it is and would therefore not be mimicking

Authors response: Thank you for your comment. We agree with your recommendation and have removed this statement .

Reviewer’s comment: Line 194 – that is more intermediate signal on T2

Authors response: We have updated the legend with the addition of “intermediate” signal.

Reviewer’s comment: Line 201 – high signal intensity where? On F? would not agree

Authors response: We have deleted this figure. Thank you for your comment.

Reviewer’s comment Line 208/209- so you are saying that all clear cell RCCs have fat in them? Reference?

Authors response: Diagnostic challenge in differentiating AML and lipid containing ccRCCs is discussed with addition of a new recently published meta-analysis reference.

Reviewer’s comment: Figure 3: C – cannot appreciate marked increase in signal as there are no precontrast images to compare to. Need either precontrast image or show subtraction images. E/F, if you want to show decrease in signal, should show ROI measurements because hard to appreciate any significant drop out. Was this lesion path proven fat poor AML?

Authors response: Since this lesion is not biopsy proven lipid poor AML. Authors choose to remove the figure from updated submission. We appreciate your comment.

Reviewer’s comment: Figure 4 – is this path proven oncocytoma? Because this figure can be of other types of solid renal mass as well based on the images provided

Authors response: Thank you for your comment about overlapping imaging appearance of hypervascular renal lesions. This renal mass was stable over three years duration favoring a benign indolent etiology such as oncocytoma rather than aggressive renal cell carcinoma. We have added comparison imaging to show interval stability.

Reviewer’s comment: Figure 5 : C and D are different mass than A and B and then you go back to the original mass in E and F.. very confusing and provides only some imaging characteristics for each mass. Also B and E are the same image, both in phase T1. Should B have been a T1 fat sat? G and H – thrombus is still enhancing on delayed phase but less so, is that what you meant? Lack of enhancement would imply dark signal.

Authors response: Thank you for your review. This figure and legend have been updated.

Reviewer’s comment: Line 296 – reference for progressive delayed enhancement? Papillary RCC tend to be hypoenhancing compared to renal parenchyma in general. https://pubs.rsna.org/doi/full/10.1148/rg.2017170039

Authors response: A new reference is added. Thank you for the comment.

Reviewer’s comment: Figure 7 C – arrowheads seem to be switched.

Authors response: We have edited the figure and legend. Thank you.

Reviewer’s comment: Line 259-261: reference?

Authors response: Thank you for your comment, appropriate references are provided.

Reviewer’s comment: Figure 9 – is this path proven because it really looks like a cyst on the limited provided imaging. What about DWI? Was the initial leiomyosarcoma so T2 hyperintense too?

Authors response:  We have updated the figure legend, stating biopsy proven leiomyosarcoma in the mesentery and new renal mass suggestive of metastasis ;  unfortunately, DWI imaging is not available for this patient.

Round 2

Reviewer 3 Report

Great revisions by the authors addressing the raised issues. There are few minor typos that need editing for example ‘clear’  misspelled in clear cell in table 1 and R erroneously capitalized in Arterial in figure 1 and missing ‘t’ in tissue on line 942.

Author Response

Reviewer 3 – Round 2:

Reviewer’s comment:

Great revisions by the authors addressing the raised issues. There are few minor typos that need editing for example ‘clear’  misspelled in clear cell in table 1 and R erroneously capitalized in Arterial in figure 1 and missing ‘t’ in tissue on line 942.

Authors response:

Again, the authors are very appreciative of detailed review, analysis and all recommendations provided by reviewer 3.

We have made the above minor revisions.